# Chemical Constituents, In Vitro Antioxidant Activity and In Silico Study on NADPH Oxidase of *Allium sativum* L. (Garlic) Essential Oil

**DOI:** 10.3390/antiox10111844

**Published:** 2021-11-20

**Authors:** Oscar Herrera-Calderon, Luz Josefina Chacaltana-Ramos, Irma Carmen Huayanca-Gutiérrez, Majed A. Algarni, Mohammed Alqarni, Gaber El-Saber Batiha

**Affiliations:** 1Department of Pharmacology, Bromatology and Toxicology, Faculty of Pharmacy and Biochemistry, Universidad Nacional Mayor de San Marcos, Jr Puno 1002, Lima 15001, Peru; 2Department of Pharmaceutical Chemical, Faculty of Pharmacy and Biochemistry, Universidad Nacional San Luis Gonzaga, Av. Los Maestros s/n, Ica 11001, Peru; luz.chacaltana@unica.edu.pe (L.J.C.-R.); irma.huayanca@unica.edu.pe (I.C.H.-G.); 3Department of Clinical Pharmacy, College of Pharmacy, Taif University, P.O. Box 11099, Taif 21944, Saudi Arabia; m.alqarni@tu.edu.sa; 4Department of Pharmaceutical Chemistry, College of Pharmacy, Taif University, P.O. Box 11099, Taif 21944, Saudi Arabia; m.aalqarni@tu.edu.sa; 5Department of Pharmacology and Therapeutics, Faculty of Veterinary Medicine, Damanhour University, Damanhour 22511, Egypt; dr_gaber_batiha@vetmed.dmu.edu.eg

**Keywords:** allyl compounds, molecular dynamic, antioxidant enzyme, oxidative stress, phytochemical study

## Abstract

*Allium sativum* L., also known as garlic, is a perennial plant widely used as a spice and also considered a medicinal herb since antiquity. The aim of this study was to determine by gas chromatography–mass spectrometry (GC–MS) the chemical profile fingerprint of the essential oil (EO) of one accession of Peruvian *A. sativum* (garlic), to evaluate its antioxidant activity and an in- silico study on NADPH oxidase activity of the volatile phytoconstituents. The antioxidant activity was tested using DPPH and β-carotene assays. An in-silico study was carried out on NADPH oxidase (PDB ID: 2CDU), as was ADMET prediction. The results indicated that diallyl trisulfide (44.21%) is the major component of the EO, followed by diallyl disulfide (22.08%), allyl methyl trisulfide (9.72%), 2-vinyl-4*H*-1,3-dithiine (4.78%), and α-bisabolol (3.32%). Furthermore, the EO showed antioxidant activity against DPPH radical (IC_50_ = 124.60 ± 2.3 µg/mL) and β-carotene bleaching (IC_50_ = 328.51 ± 2.0). The best docking score on NADPH oxidase corresponds to α-bisabolol (ΔG = −10.62 kcal/mol), followed by 5-methyl-1,2,3,4-tetrathiane (ΔG = −9.33 kcal/mol). Additionally, the volatile components could be linked to the observed antioxidant activity, leading to potential inhibitors of NADPH oxidase.

## 1. Introduction

Essential oils (EOs) are natural products obtained from aromatic plants and can be extracted from leaves, roots, stems, flowers, and seeds, among others [1]. EOs are widely used in the food, cosmetic, alternative therapy (such as aromatherapy) and pharmaceuticals industries [2]. In terms of volatile chemical composition, EOs are mainly constituted by monoterpenes, sesquiterpenes, phenols, and alcohols. However, allyl structures and phenylpropanoids also constitute the phytochemical profile of some EOs [3]. EOs have been linked to antioxidant activity [4] as free radical scavengers and metal chelators [5], as well as to other biological activities such as anti-inflammatory, analgesic [6], sedative [7], antibacterial [8], antiviral [9], neuroprotective [10], and antifungal properties [11].

The generation of free radicals leads to oxidative stress in cells, which can trigger aging and degenerative diseases. The exposure to environmental stress, UV-radiation, viral and bacterial infection and carcinogenic chemicals, among other things, might also cause severe damage, brought about by the lipid peroxidation of polyunsaturated fatty acids and, consequently, the liberation of toxic metabolites [12]. Studies have revealed that essential oils inhibit lipid peroxidation in foods and in the biochemical process related to oxidative stress [2]. According to Amorati et al., the chemical fingerprint of EOs is mainly comprised of terpenoid and phenylpropanoid compounds [13]. Monoterpenes, free phenols, and allylic alcohols have all demonstrated potential antioxidant capacity in oxidative stress models, while other compounds grouped as sesquiterpenes, and non-isoprenoids have shown low antioxidant activity [14].

Nowadays, computational models such as molecular docking are carried out as a bioinformatic tool to study the inhibition of several enzymes that negatively affect the antioxidant activity, such as xanthine oxidase, nitric oxide synthases (NOS), cytochrome P450 reductase, and nicotinamide-adenine dinucleotide phosphate (NADPH) oxidase, as well as the mitochondrial electron transport chain. Furthermore, it is known that reactive oxygen species (ROS) generation occurs in the mitochondria via oxidative phosphorylation and through the enzyme NADPH oxidase [15]. Additionally, NADPH oxidase-mediated cell proliferation participates in intracellular signaling processes and has been observed in a variety of cancer cells and in tissue repair processes [16]. Thus, the screening of inhibitory molecules on this target could be useful to ameliorate numerous chronical or degenerative diseases and to find selective and non-toxic inhibitors of NADPH oxidases, providing new drugs for the treatment of diseases related to oxidative stress-dependent processes [17].

*Allium sativum* L. (family Amaryllidaceae), also known as garlic is widespread across the world, being used mainly in gastronomy and for its medicinal properties. Within its medical uses, several pharmacological activities have been evaluated, such as its potential anticancer, hypotensive, hepatoprotective, hypoglycemic, antimicrobial, and immunomodulatory effects [18]. Garlic essential oil contains sulfur compounds as the main volatile phytochemicals, dominated by allyl polysulfides, such as diallyl disulfide, diallyl trisulfide, and diallyl disulfide. Different methods to obtain EOs adopted in some studies which might affect the chemical composition due to enzymatic biotransformation processes occurring during the extraction process, i.e., hydrodistillation compared to the microwave-assisted extraction method [19,20]. The type of apparatus used during the extraction could also influence the results, i.e., use of a Clevenger apparatus vs. industrial extraction. Considering the role of the edaphic factor on the plant growth and chemical composition of different metabolites of interest, the chemical profile of essential oils has been shown to be affected by all these factors [21]. Altitude also plays an important role; to date only the results for garlic collected in low altitudes ranging between 500 and 1000 m.a.s.l. have been reported [22]. Additionally, remarkable qualitative and quantitative differences have been found in EOs extracted after following different drying procedures [23]. In Peru, garlic is used mainly as a food additive, and according to National Institute of Agrarian Innovation (INIA-PERU), Peru has several accessions of garlic such as Purple, Criollo or Napurí, Barranquino, Massone, Pata de Perro, and White Chinese [24], with White Chinese being one of the most consumed and commercialized accessions.

To date, some accounts of the antioxidant capacity of garlic EOs determined by different methods have been reported. However, those garlic samples were cultivated at low altitudes and their chemical compositions could differ to those of garlic cultivated in high altitude zones. In-silico studies allow one to identify molecules with promising inhibitory effects on any antioxidant target such as NADPH oxidase, which is considered an important antioxidant marker in biological systems. The focus on antioxidant phytochemicals contained in EOs is directly linked to their application for the prevention of oxidative damage caused by ROS. Hence, low-molecular antioxidants such as allyl polysulfide structures enhance organism stability under conditions of oxidative stress. Based on all these antecedents, the aims of this research were: (1) To determine the phytochemical constituents of garlic EO by gas chromatography–mass spectrometry (GC–MS) and their antioxidant activity against the 1,1-diphenyl-2-picrylhydrazyl (DPPH) radical and β-carotene bleaching; (2) to evaluate in-silico the inhibitory effect of the volatile phytochemicals of the essential oil from *A. sativum* L. (garlic) on NADPH oxidase (PDB ID: 2CDU).

## 2. Materials and Methods

### 2.1. Chemicals

All analytical grade (99.5%) solvents (dichloromethane, chloroform, methanol) and hydrogen peroxide were purchased from Merck (Darmstadt, Germany). 2,2-Diphenyl-1-picrylhydrazyl (DPPH), β-carotene, linoleic acid, Trolox, and rutin were purchased from Sigma Aldrich (St Louis, MO, USA).

### 2.2. Plant Material

A quantity of 800 g of *A. sativum* (bulbs) cultivated in the Arequipa region of Peru (2.335 m.a.s.l.), was used to carry out the experimental procedures. Bulbs were cleaned and peeled before being placed in a Clevenger equipment to obtain the essential oil by hydrodistillation for 2 h [25]. The essential oil was separated by decantation, then anhydrous Na_2_SO_4_ was added to eliminate any remaining water drops. Finally, the EO was stored in a sealed amber vial until further use.

### 2.3. Identification of Volatile Compounds by Gas Chromatography–Mass Spectrometry (GC–MS)

Volatile chemicals were determined with a GC–MS system (7890 Gas Detector and 5975C Mass Spectrometer Detector, Agilent Technologies, Santa Clara, CA, USA). Then, 0.0136 g of the sample was weighed and mixed with 0.5 mL of dichloromethane. Next, 1.0 µL of the working solution was injected into the equipment in splitless mode (split ratio: 20:1). The EO was run on a J&W 122-1545.67659 DB-5ms column (60 m × 250 μm × 0.25 μm, Agilent Technologies). The working conditions were as follows: the temperature program started at 40 °C, with increments of 5 °C/min up to 180 °C, followed by increases of 2.5 °C/min up to 200 °C for 5 min and finally 10 °C/min up to 300 °C, followed by holding for 3 min. The helium flow rate was at 1 mL/min. Volatile chemicals were identified and confirmed by comparing the mass spectrum of the compounds with the NIST20 library data [26].

### 2.4. Antioxidant Activity

#### 2.4.1. 2,2-Diphenyl-1-picrylhydrazyl (DPPH) Assay

The 2,2-diphenyl-1-picrylhydrazyl (DPPH) method according to Rojas-Armas et al. [27] with slight modifications, was used as an organic radical activity assay. For 0.1 mM of the DPPH solution, methanol was used as a solvent, and 100 µL of this solution was mixed with 900 µL of EO at different concentrations (0–1000 µg/mL). Then, the reaction tubes were incubated at room temperature for 30 min and protected from light. Finally, the absorbance was measured at 517 nm using a Genesys 20 spectrophotometer (Thermo Scientific, Waltham, MA, USA). Trolox was used as an antioxidant control. The inhibitory concentration (IC_50_) was calculated from the plot of inhibition percentage against the sample concentration.

#### 2.4.2. β-Carotene Bleaching Assay

The lipid peroxidation activity was determined by the β-carotene bleaching method according to Tepe et al. [28]. A stock solution containing 0.5 mg of β-carotene in 1 mL of chloroform was then mixed with 200 mg of Tween 40 and 25 μL of linoleic acid. Then, the chloroform was removed under vacuum at 40 °C for 5 min using a R-100^®^ rotary evaporator (Buchi, Flawil, Switzerland). Subsequently, 100 mL of 3% aqueous hydrogen peroxide solution was added to the residue and mixed for 5 min until an emulsion was obtained. Then, aliquots (2.5 mL) of the emulsion were mixed with garlic EO (350 μL) and rutin. All test tubes were incubated at 25 °C up to 48 h and read at 490 nm using a Genesys 20 spectrophotometer.

### 2.5. Molecular Docking of the Phytochemical Volatiles of Garlic on the Receptor NADPH Oxidase (PDB ID: 2CDU)

Prior to the docking study, 13 volatile oil compounds of *Allium sativum* (garlic) were drawn in ChemDraw 19.0 (Perkin Elmer, Waltham, MA, USA), and subsequently, the geometry was optimized. The receptor NADPH oxidase was retrieved from the protein data bank (PDB ID: 2CDU). Before performing molecular interaction studies, NADPH oxidase was further curated for missing side-chain residues using the openMM tool. Molecular docking studies were performed with Autodock v 4.2.6 (The Scripps Research Institute, La Jolla, CA, USA). The binding cavity for ligand compound docking in NADPH oxidase was determined from the predefined co-crystallized X-ray structure from RCSB PDB. The residue positions were calculated within 3 Å from the co-crystallized ligand. After the cavity selection in each case, the co-crystallized ligands were removed using the Chimera tool (https://www.cgl.ucsf.edu/chimera/, accessed on 11 September 2021), and subsequently, the energy was minimized using the steepest descent and conjugate gradient algorithm. Then, finally, merging the nonpolar hydrogens, both the receptor and target compound were saved in pdbqt format. A grid box was created with parameters X = 68, Y = 58, and Z = 64 Å with 0.3 Å spacing. Following the Lamarckian genetic algorithm (LGA), docking studies of the protein–ligand complex were performed to achieve the lowest free energy of binding (∆G). During the molecular docking studies, three replicates were performed, with a total number of 50 solutions computed in each case, with a population size of 500, a number of evaluations of 2,500,000, a maximum number of generations of 27,000, and rest the default parameters were allowed. After docking, the RMSD clustering maps were obtained by the re-clustering command with a clustering tolerance of 0.25, 0.5, and 1 Å, respectively, in order to obtain the best cluster with the lowest energy score and a high number of populations.

### 2.6. Molecular Dynamic (MD) Studies

The MD simulation studies were carried on the best docked complexes for NADPH oxidase with α-bisabolol and standard diallyl trisulfide using Desmond 2020.1 from Schrödinger, LLC (New York, NY, USA). The OPLS-2005 force field and explicit solvent model with the SPC water molecules were used in this system. Na+ ions were added to neutralize the charge. Moreover, 0.15 M, of the NaCl solution was added to the system to simulate the physiological environment. The NPT ensemble was set up using the Nose–Hoover chain coupling scheme with a temperature of 27 °C, a relaxation time of 1.0 ps, and a pressure of 1 bar, maintained in all of the simulations. A time step of 2 fs was used. The Martyna–Tuckerman–Klein chain coupling scheme barostat method was used for pressure control with a relaxation time of 2 ps. The particle mesh Ewald method was used for calculating long-range electrostatic interactions, and the radius for the coulomb interactions were fixed at 9Å. The RESPA integrator was used for a time step of 2 fs for each trajectory calculating the bonded forces. The root means square deviation (RMSD), radius of gyration (Rg), root mean square fluctuation (RMSF), and number of hydrogen (H-bonds) were calculated to monitor the stability of the MD simulations [29].

### 2.7. ADMET Prediction

The absorption, distribution, metabolism, excretion, and toxicity (ADMET) properties of the compounds determined in the essential oil of garlic were calculated using the pkCSM online tool (http://biosig.unimelb.edu.au/pkcsm/prediction, accessed on 21 October 2021) [30] and SwissADME (http://www.swissadme.ch/, accessed on 21 October 2021) [31].

### 2.8. Statistical Analysis

The preparation of essential oil was carried out three times, and the mean and SD of the three independent experiments are presented. The antioxidant assays were repeated three times. The statistical tools employed were Student’s *t*-test, two-tailed, and the IC_50_ values were estimated by linear regression statistics. *p*-Values less than 0.05 were considered statistically significant. GraphPad Prism program version 6.0 (GraphPad Software, La Jolla, CA, USA) was used to develop the statistical analysis.

## 3. Results and Discussion

### 3.1. Chemical Profile of the Essential Oil of A. sativum

The obtained EO showed a pale yellow color, an extraction yield of 0.78% (v/dry weight), and a density of 0.95 ± 0.01 g/mL at 20 °C. The volatile components of garlic essential oil were analyzed by GC–MS and are presented in Table 1. According to our results, we identified 17 compounds, four of which are of unknown structures, which accounted for 100% of the total composition. The analysis identified diallyl trisulfide (retention time 24.97 min) as the main component (44.21%) of the volatile chemicals, followed by diallyl disulfide (22.08%), allyl methyl trisulfide (9.72%), and 2-vinyl-4*H*-1,3-dithiine (4.78%). According to Appendix A, the total time for the evaluation was 50 min. 

Regarding the phytochemical analysis of garlic essential oil by GC–MS, our results differ to those of other investigations and could be explained by the different extraction methods used to obtain the EO, such as conventional or non-conventional techniques. For example, in sono-hydrodistillation and ultrasound-assisted hydrodistillation, the obtained garlic essential oil has, as a major component, diallyl disulfide, whilst is lower in hydrodistillation [32]. In our study, the most representative molecule was diallyl trisulfide, with 44.21%, followed by diallyl sulfide, with 22.08%. However, in recent studies, as reported by Thuy et al., allyl disulfide was the main component, with 28.44% [9]. In an analysis of garlic EO from Cameroon, diallyl trisulfide (41.62%) and diallyl disulfide (19.74%) were the major components [33]. In a Tunisian study, allyl disulfide (28.0%) and eugenol (15.37%) were the most abundant constituents [34]. In Saudi Arabia, the major component was allyl methyl trisulfide, with 13.10%. [35]. Garlics from Thailand showed values of diallyl disulfide, diallyl trisulfide, and diallyl tetrasulfide, equivalent to 31.67%, 31.56%, and 13.48%, respectively [36]. On the contrary, the drying procedure also affects the composition of sulfur components, i.e., garlic EO from Tunisia obtained by freeze drying contained 45.9% diallyl trisulfide compared to 42.3% for an oven-dried extract and 37.3% for an air-dried extract [37]. The diversity of the chemical composition in garlic EO might be related with external factors such as temperature, soil composition, climate conditions, environmental stress, ecosystem, and altitude. Arequipa is situated at 2335 m.a.s.l., which could be an advantage to produce an EO with a phytochemical profile different to other varieties of garlic found across the world [38].

### 3.2. Antioxidant Profile of A. sativum Essential Oil

Garlic EO exhibited strong antioxidant activity, as shown in Table 2. Trolox showed better antioxidant activity than the EO. On the contrary, there was a significant difference between EO and Trolox concentrations (*p* = 0.0002). The EO of *A. sativum* showed a good antioxidant response, but other research has reported different values; for example, according to Lawrence et al., the EO of garlic grown in the north Indian plains showed an IC_50_ value of 0.5 mg/mL [39]. Another IC_50_ value that has been reported is 7.67 mg/mL, for an oil obtained by hydrodistillation [40]. In our study the EO showed an IC_50_ of 124.60 ± 2.5 µg/mL, which is different to the findings of Ndoye et al. [33], with an IC_50_ value of 0.19 µg/mL. This is a high value compared to the other results, which could be due to the presence of diallyl trisulfide, diallyl disulfide, and methylallyl di- and trisulfides. Although diallyl sulfide was the main component of the garlic EO produced by hydrodistillation, dialyl disulfide and allyl methyl sulfide did not demonstrate any antioxidant action when tested as inhibitors of the controlled autoxidation of isopropylbenzene or styrene, implying that they are oxidized together with the oxidizable substrate [13]. We detected diallyl trisulfide as the major compound, whilst other sulfur volatiles have been shown to be abundant in other garlic EOs. However, some techniques to extract essential oil might affect its antioxidant activity, according to Boubechiche et al. [32]. The EO obtained by hydrodistillation had a better antioxidant capacity (IC_50_ = 0.96 mg/mL) than that obtained using the ultrasound-assisted (IC_50_ = 1.176 mg/mL) and sono-hydrodistillation processes (IC_50_ = 1.234 mg/mL). After microwave-assisted extraction, the EO showed an inhibition percentage of 72.06% at 500 μg/mL [41]. Factors such as altitude, climate conditions, and chemotype varieties may also be responsible for the differences observed in our study [42]. Regarding the β-carotene bleaching assay, there was a significant difference between rutin and the garlic EO (*p* = 0.0012), being the antioxidant standard more than garlic EO. In a recent study by Ncir et al. [34], an IC_50_ value of β-carotene equivalent to 0.2 ± 0.02 mg/mL was reported, similar to our results but contrary to the DPPH assay, with an IC_50_ value of 0.048 ± 0.007 mg/mL.

Another factor considered was the presence of α-bisabolol, which has not been found in other garlic EOs and which could influence the antioxidant activity, as shown in Table 2 in comparison to other garlic EOs from across the world.

### 3.3. Molecular Docking of the Phytochemical Constituents of the Essential Oil from A. sativum

Molecular docking studies were carried out in order to understand the interaction profile of various volatile oil compounds present in *A. sativum* with NADPH oxidase. Out of the 13 specific compounds found abundant in the chromatography results, α-bisabolol displayed lowest the binding energy, with (ΔG) −10.62 kcal/mol and a predicted inhibitory concentration (Ki) of 0.14 µM (Table 3). The principal residues of NADPH oxidase LYS187 and TYR188 were involved in conventional polar hydrogen bond formation with α-bisabolol (Figure 1). On the contrary, 5-methyl-1,2,3,4-tetrathiane and 4*H*-1,2,3-trithiine also displayed better binding with NADPH oxidase, followed by α-bisabolol, with ΔG −9.33 and −9.05 kcal/mol, respectively (Table 1). However, the predicted Ki was observed to be 1.24 and 1.90 µM, respectively (Table 1). The common residue of NADPH oxidase CYS133 was involved in both the cases, forming polar hydrogen bonds (Figure 1), while other than CYS133, GLY244 also indulged in forming conventional hydrogen bonds with 5-methyl-1,2,3,4-tetrathiane (Figure 1). All of the other ligands’ interaction profiles are presented in Table 3 and Appendix A.

The inhibition of NADPH oxidase in silico by sulfur components and alpha-bisabolol could be correlated with the in vitro results presented by Schepetkin et al., including the garlic EO and three compounds (diallyl trisulfide, ajoene, and allicin), which inhibited the neutrophil ROS production, with diallyl trisulfide being the major component attributed to ROS inhibition [43]. Additionally, S-allylcysteine showed antioxidant activity, inhibiting gp91^phox^ and gp22^phox^ of NADPH oxidase, where gp91^phox^ was the catalytic subunit and gp22^phox^ is a membrane protein, both of which formed a complex generating superoxide radical, and its over-activation has been linked to several renal diseases [44].

### 3.4. Molecular Dynamics of the Phytoconstituents of the Essential Oil from A. sativum

In molecular dynamics, the two most common parameters of structural fluctuations are the root mean square deviation (RMSD) and the root mean square fluctuations (RMSF). The RMSD measures the average displacement of the atoms at an instant of the simulation relative to a reference structure, generally the first frame of the crystallographic structure or simulation. The RMSF measures the displacement of a particular atom, or group of atoms, relative to the reference structure, averaged over the number of atoms. The RMSD is useful for analysis of time-dependent motions of a structure. It is frequently used to discern whether a structure is stable during the time-scale of the simulations [45].

Molecular dynamics studies of NADPH oxidase with α-bisabolol and diallyl trisulfide were carried out for a 100 ns simulation time scale. A total of 100 ns of simulation time analysis of the trajectories displayed convergence of the root mean square deviation (RMSD) for α-bisabolol with an average deviation 0.1 Å (Figure 1A, dark cyan). Meanwhile, with diallyl trisulfide, the RMSD showed little more fluctuation compared to alpha-bisabolol, with an average deviation 0.5 Å (Figure 2A, orange). RMSD deviations of the bound complexes were exhibited within an acceptable range, and with alpha-bisabolol, the Cα of NADPH oxidase was more stable. The RMSF of individual amino residues of NADPH oxidase over the function of a 100 ns time scale displayed low fluctuations in α-bisabolol, with an average of 0.1 Å (Figure 2B, drak cyan). Low fluctuations of the amino acid residues indicate higher stability from a converged structure. However, amino acid residues of NADPH oxidase bound to diallyl trisulfide fluctuated more at positions 70 and 370, respectively, as compared to the α-bisabolol-bound complex (Figure 2B, orange).

The radius of gyration is the measure of compactness of the protein in the ligand-bound state. The MD simulation of 100 ns of NADPH oxidase Cα atoms complexed with α-bisabolol displayed a slight lowering of the trajectory due to compactness of the complex (Figure 3A, dark cyan); meanwhile, the complex with diallyl trisulfide (orange) displayed more fluctuations and a less compact structure compared to the α-bisabolol-bound complex (Figure 3A, orange).

The number of hydrogen bonds formed between proteins and ligands is an important factor when analyzing a stable complex throughout a simulation. Here, in this case, the number of H-bonds formed between NADPH oxidase and α-bisabolol displayed a consistent interaction, with an average of 1.5 throughout the 100 ns simulation (Figure 2B). In contrast, with diallyl trisulfide, no hydrogen bond formations were recorded. The total energy of the system is another essential function to draw a conclusion regarding the stability of a complex. NADPH oxidase-α-bisabolol displayed an average of −35 kcal/mol, where most of the energies were favorably contributed by the non-bonded Coulomb and van der Waal’s forces (vdW) interactions of −5 and −25 kcal/mol, respectively (Figure 4A). On the contrary, the diallyl trisulfide-bound NADPH oxidase complex displayed an average energy of −27 kcal/mol, where the vdW energy contributed more favorably than the Coulomb energy (Figure 4B). The comparative energy plots indicate a more stable and converged structure of α-bisabolol-bound NADPH oxidase than the standard diallyl trisulfide. Therefore, from the simulation studies, it can be suggested that α-bisabolol has the potential to be a better inhibitor against NADPH oxidase and can substitute the application of diallyl trisulfide.

### 3.5. ADMET Profiles

The chemical constituents of the garlic essential oil were analyzed using the online pkCSM tool to predict the absorption, distribution, metabolism, excretion, and toxicity profiles. The results revealed that all compounds had a molecular weight ranging between 100 and 400 g/mol, which is important for penetrability, because the profile to achieve this parameter is for those compounds with values less than 500 g/mol. The Caco-2 permeability had values above 0.90 and a high intestinal absorption (90–95%), which would predict that all compounds of the garlic EO will be absorbed in the small intestine [29]. Due to the lipophilic character of chemical components in garlic EOs, they tend to form micelles and are digested in the small intestine [46]. The skin permeability ranged from −1.29 to −2.232 cm/h (≤2.5), which means that volatile phytochemicals easily penetrate the skin.

The volume distribution (VDss) is acceptable due to its values above −0.15. On the contrary, 11 compounds are probably able to penetrate the blood–brain barrier (BBB) (log BB > 0.3), but 3H-1,2-dithiole and 3-vinyl-1,2-dithiacyclohex-4-ene had a medium value under 0.3, which would result in difficultly accessing the brain. The central nervous system (CNS) permeability achieved between −2.644 and −1.649; therefore, only 5-methyl-1,2,3,4-tetrathiane is able to permeate the central nervous system (logP > −2), as values logP < −3 are unable to penetrate it. Table 4 shows that all of constituents are not able to interfere with CYP2D6 and CYP3A4 and would not be metabolized by either. The excretion parameters showed that the total clearance (0.103 and 1.363 log mL/min/kg) achieved positive values, meaning a quick excretion [15]. Volatile organic compounds can influence the expression and activity of cytochrome P450 enzymes and transferases involved in the metabolism of other drugs in the system. Additionally, drug interactions of essential oil components should be considered if any medication is used together with EOs. Furthermore, all compounds showed no potential contraindication, because they are not substrates of the organic cation transport 2 (OCT2), which is involved in the uptake and secretion of cationic drugs. Regarding the toxicity, the acute oral toxicity in rats (LD_50_) ranged from 1.739 to 3.035 mol/kg, meaning a low toxicity. Additionally, they were not considered hepatotoxic substances (Table 4).

The main limitation of this study was in the development of the in vitro and in silico study to demonstrate the antioxidant capacity of the garlic EO, being very important for in vivo models, i.e., mammalian cell lines, as well as for experimental animals. However, these findings pretend to show the probable mechanism of molecular interaction of the volatile components of the EO on NADPH oxidase. In addition, there lacked a comparison of the results determined in the DPPH and β-carotene bleaching assays with other garlic EOs studied elsewhere in the world. Furthermore, there were limitations in showing the chemical profile of Peruvian garlic, which is cultivated in different lands affected by altitude, climate, and temperature, among others. Garlic EO might be a promissory antioxidant agent used in phytotherapy or in the food industry. Furthermore, the chemical structures such as α-bisabolol and sulfur components determined in the EO could be used as templates for the design of new drugs to inhibit NADPH oxidase in the future.

## 4. Conclusions

We concluded that the essential oil of *A. sativum* cultivated in Peru at 2335 m.a.s.l. has a low percentage yield, and analysis with GC–MS revealed the presence of diallyl trisulfide, with 44.23%, as the main volatile component of the EO, followed by 14 compounds of sulfur structures, with 55.02%, and two oxygenated terpenes, namely, α-bisabolol and one unknown constituent with the formula C_22_H_42_O_4_, which represent 3.75%.

Regarding the antioxidant capacity in vitro, two assay methods were carried out, namely, DPPH and β-carotene bleaching assays, showing a good inhibitory capacity over other types of garlic EOs reported across the world. The presence of two compounds not detected in other garlic EOs across the world, such as α-bisabolol (a sesquiterpene), would be vital in maintaining their high antioxidant power. However, garlic EO were less active than the standard antioxidants like Trolox and rutin. In the in-silico study, the activity of the phytoconstituents of the garlic EO on NADPH oxidase were studied, being α-bisabolol the compound with the best docking score. Furthermore, this compound showed good stability during the molecular dynamics study carried out across 100 ns, whilst diallyl trisulfide was not stable, demonstrating that those ligands with low-energy docking scores have poor stability on the target protein. Additionally, according to the ADMET prediction, all garlic components can be absorbed by oral and transdermal administration, which represent a great advantage against other compounds that experience difficulty in penetrating the gastrointestinal barrier. No toxicity was predicted, but further studies have to be developed to confirm these findings using in vivo models.

## Figures and Tables

**Figure 1 antioxidants-10-01844-f001:**
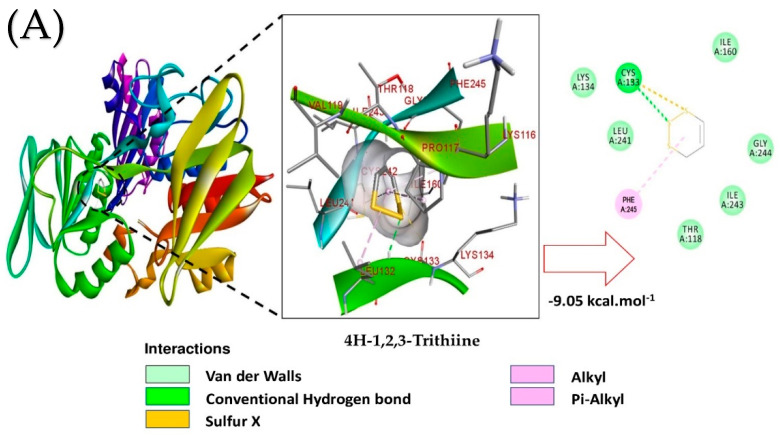
Molecular interaction studies of the most active phytochemical constituents of garlic essential oil with NADPH oxidase (PDB ID: 2CDU). (**A**): 4H-1,2,3-trithiine, (**B**): 5-methyl-1,2,3,4-tetrathiane, and (**C**): α-bisabolol. Surface view (left panel), and 2D (right panel) interactions.

**Figure 2 antioxidants-10-01844-f002:**
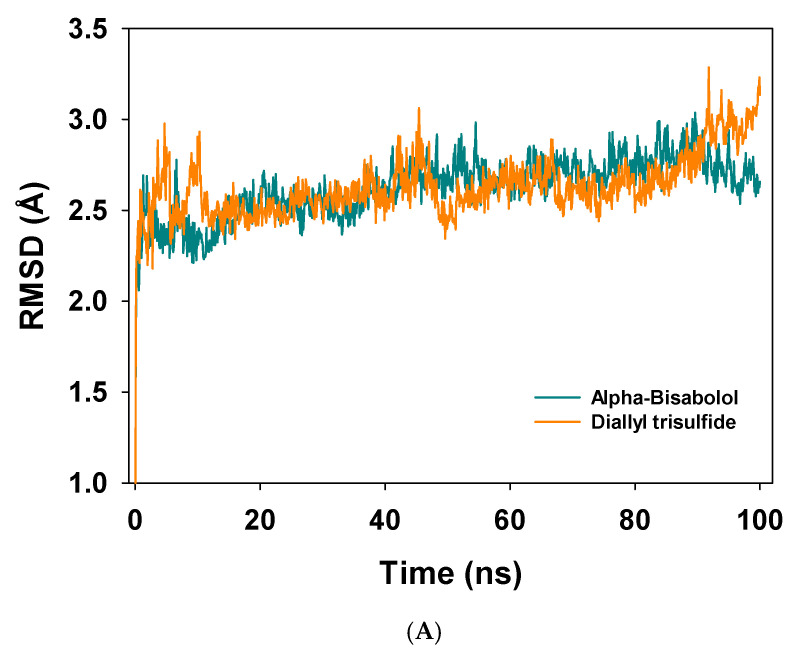
(**A**) RMSD plot of Cα atoms of NADPH oxidase displaying deviations from the mean with α-bisabolol (dark cyan) and diallyl trisulfide (orange) over a 100 ns time scale simulation. (**B**) RMSF plot of amino acids of NADPH oxidase bound to α-bisabolol (dark cyan) and diallyl trisulfide (orange) over a 100 ns time scale simulation.

**Figure 3 antioxidants-10-01844-f003:**
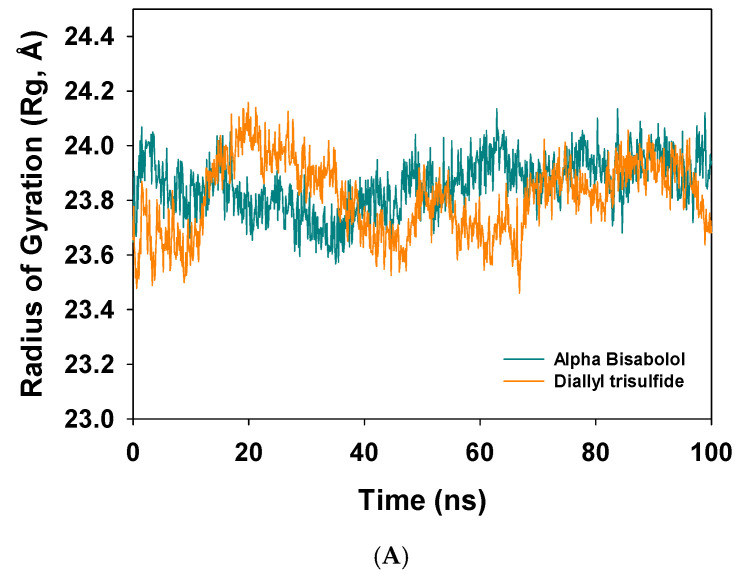
(**A**) Radius of gyration (Rg) plot of Cα atoms of NADPH oxidase complexed with α-bisabolol (dark cyan) and diallyl trisulfide (orange). (**B**) Number of hydrogen bonds formed between NADPH oxidase and α-bisabolol throughout a simulation time of 100 ns.

**Figure 4 antioxidants-10-01844-f004:**
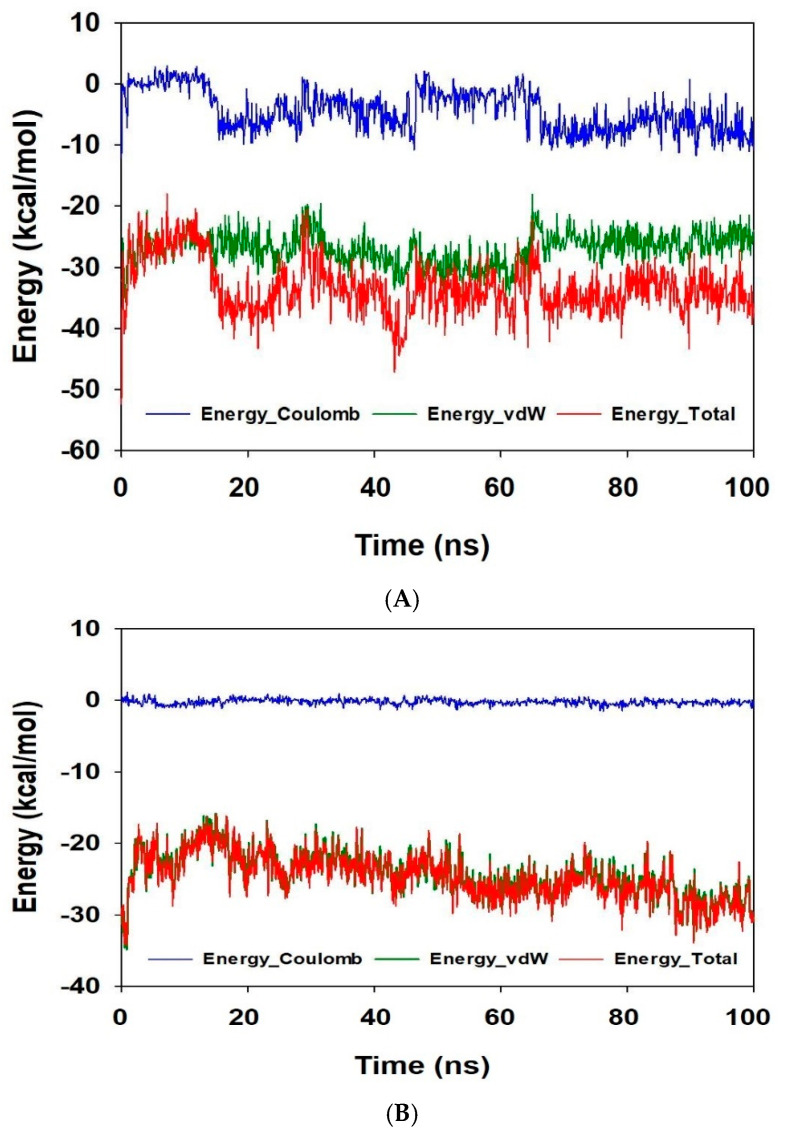
Energy plots of NADPH oxidase complexed with (**A**) α-bisabolol and (**B**) diallyl trisulfide.

**Table 1 antioxidants-10-01844-t001:** Chemical composition of the volatile oil of *A. sativum* (garlic).

N°	Compound Name (NIST08.L)	Rt(min)	Molecular Formula/Molecular Mass	% In Sample(Relative Areas)	Chemical Structure
1	Allyl methyl disulfide	12.75	C_4_H_8_S_2_(120.36)	0.47 ± 0.01	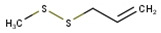
2	3*H*-1,2-Dithiole	14.18	C_3_H_4_S_2_(104.19)	2.41 ± 0.02	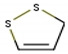
3	Diallyl disulfide	18.11	C_6_H_10_S_2_(146.3)	22.08 ± 0.11	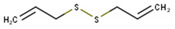
4	1-Propenyl 2-propenyl-(*E*)-disulfide	18.69	C_6_H_10_S_2_(146.27)	0.92 ± 0.01	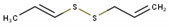
5	Allyl methyl trisulfide	20.06	C_4_H_8_S_3_(152.29)	9.72 ± 0.05	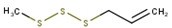
6	Unknown	20.92	C_6_H_10_S_2_(146.27)	1.08 ± 0.01	n.d.
7	3-Vinyl-1,2-dithiacyclohex-4-ene	21.71	C_6_H_8_S_2_(144.25)	2.56 ± 0.01	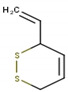
8	4H-1,2,3-Trithiine	22.18	C_3_H_4_S_3_(136.25)	3.07 ± 0.01	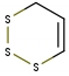
9	2-Vinyl-4*H*-1,3-dithiine	22.46	C_6_H_8_S_2_(144.25)	4.78 ± 0.01	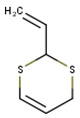
10	Unknown	24.66	C_9_H_16_O_2_S(188.29)	0.50 ± 0.01	n.d.
11	Diallyl trisulfide	24.97	C_6_H_10_S_3_(178.33)	44.21 ± 0.08	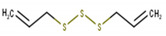
12	1-Allyl-3-propyltrisulfane	25.22	C_6_H_12_S_3_(180.34)	1.37 ± 0.01	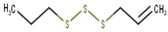
13	5-Methyl-1,2,3,4-tetrathiane	27.20	C_3_H_6_S_4_(170.32)	1.55 ± 0.01	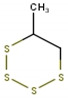
14	Unknown	31.50	C_9_H_16_S_3_(220.42)	0.85 ± 0.01	n.d.
15	Diallyl tetrasulfide	31.55	C_6_H_10_S_4_(210.39)	0.68 ± 0.01	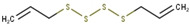
16	α-Bisabolol	35.42	C_15_H_26_O(222.37)	3.32 ± 0.03	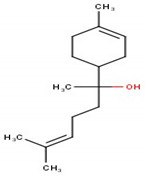
17	Unknown	45.26	C_22_H_42_O_4_(370.57)	0.43 ± 0.01	n.d.

Rt, retention time; n.d., not determined.

**Table 2 antioxidants-10-01844-t002:** Antioxidant activities of the *A. sativum* essential oil.

Samples	Antioxidant Activity	
	DPPH IC_50_ (μg/mL)	β-Carotene IC_50_ (μg/mL)
Essential oil of *Allium sativum*	124.60 ± 2.3 **	328.51 ± 2.0 *
Trolox	0.54 ± 0.02	-
Rutin	-	3.5 ± 0.05

Values are reported as the mean ± SD of three experiments. * *p* < 0.01; ** *p* < 0.001. Student’s *t*-test, two-tailed.

**Table 3 antioxidants-10-01844-t003:** Ligand interaction energies and inhibitory concentrations with NADPH oxidase in the molecular docking study.

No.	Ligand Name	ΔG (kcal/mol)	Ki (µM)	Residues Involved in Polar Bonds
1	Allyl methyl disulfide	−6.85	15.0	VAL214
2	3*H*-1,2-Dithiole	−7.45	5.45	CYS133
3	Diallyl disulfide	−7.19	8.48	ILE243, GLY244
4	1-Propenyl 2-propenyl-(*E*)-disulfide	−6.59	23.2	LYS187
5	Allyl methyl trisulfide	−6.65	21.1	MET33
6	3-Vinyl-1,2-dithiacyclohex-4-ene	−8.87	2.78	VAL81
7	4*H*-1,2,3-Trithiine	−9.05	1.90	CYS133
8	2-Vinyl-4*H*-1,3-dithiine	−7.69	3.64	VAL81
9	Diallyl trisulfide	−7.36	6.37	GLY244
10	1-Allyl-3-propyltrisulfan	−7.76	3.32	GLY244
11	5-Methyl-1,2,3,4-tetrathiane	−9.33	1.24	CYS133, GLY244
12	Diallyl tetrasulfide	−6.21	44.4	-
13	α-Bisabolol	–10.62	0.140	LYS187, TYR188

VAL, valine; CYS, cysteine; ILE, isoleucine; GLY, glycine; LYS, lysine; MET, methionine; TYR, tyrosine.

**Table 4 antioxidants-10-01844-t004:** ADMET properties of the chemical constituents of the *A. sativum* essential oil (in the metabolism analysis for CYP 2D6 inhibitors, CYP 3A4 inhibitors, renal OCT2 substrate, and hepatotoxicity, no data were obtained).

N°	Absorption			Distribution			Excretion	Toxicity	
	Caco-2 (Log Papp in 10^−6^ cm/s)	Intestinal Absorption (% Absorbed)	Skin Permeability (Log Kp)	VDss (Log L/kg)	BBB Permeability (Log BB)	CNS Permeability (Log PS)	Total Clearance (Log mL/min/kg)	Oral Rat Acute Toxicity (LD_50_ = mol/kg)	Oral Rat Chronic Toxicity (LOAEL = Log mg/kg_bw/day)
1	1.394	94.604	−1.761	0.098	0.332	−2.336	0.444	2.512	1.726
2	1.384	95.015	−2.232	0.159	0.183	−2.599	0.287	2.674	1.733
3	1.406	94.007	−1.374	0.197	0.743	−2.157	0.366	2.433	2.026
4	1.397	92.885	−1.652	0.112	0.437	−2.435	0.347	2.845	1.728
5	1.398	94.877	−1.865	0.241	0.377	−2.439	0.496	2.447	1.777
6	1.394	93.326	−2.088	0.148	0.27	−2.644	0.161	2.956	1.756
7	1.398	94.908	−1.865	0.241	0.378	−2.439	0.467	2.442	1.752
8	1.403	92.573	−1.449	0.216	0.767	−2.309	0.446	2.711	1.857
9	1.403	92.573	−1.449	0.216	0.767	−2.309	0.446	2.711	1.857
10	1.403	92.097	−1.425	0.225	0.757	−2.309	0.389	2.75	1.931
11	1.427	91.045	−1.29	0.165	0.693	−1.649	0.103	2.863	1.812
12	1.406	90.609	−1.552	0.224	0.759	−2.402	0.336	3.035	1.843
13	1.505	93.014	–1.761	0.42	0.605	–2.541	1.363	1.739	1.178

1: Allyl methyl disulfide; 2: 3*H*-1,2-dithiole; 3: diallyl disulfide; 4: 1-propenyl-2-propenyl-(*E*)-disulfide; 5: allyl methyl trisulfide; 6: 3-vinyl-1,2-dithiacyclohex-4-ene; 7: 4*H*-1,2,3-trithiine; 8: 2-vinyl-4*H*-1,3-dithiine; 9: diallyl trisulfide; 10: 1-allyl-3-propyltrisulfane; 11: 5-methyl-1,2,3,4-tetrathiane; 12: diallyl tetrasulfide; 13: α-bisabolol.

## Data Availability

The data is contained within the article and Appendix A.

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
