# Peer review of "Chemical Constituents, In Vitro Antioxidant Activity and In Silico Study on NADPH Oxidase of Allium sativum L. (Garlic) Essential Oil"

_antioxidants, 2021, doi:10.3390/antiox10111844_

Round 1
Reviewer 1 Report
The manuscript described the essential oil (EO) of A. sativum cultivated in Peru. According to the GC-Ms, the main components of the EO has been identified to be Diallyl trisulfide which exhibited antioxidant activity against DPPH radical and β-carotene bleaching. Moreover, the computational studies have revealed the best docking score for α-Bisabolol on NADPH oxidase. The manuscript is interesting, and can be accepted after the following minor revision.
- Line168-169, Page 4, the authors identified 17 compounds, and four of them are unknown structures. Could the authors identify the specific structure of the four unknown compounds?
- Line 181-188, page 4, the authors have stated diversity of chemical composition in garlic EO. Will differences in composition lead to differences in application of the garlic oil?
- Some typing mistakes should be revised through out the manuscript. For example, Line 170, “the analysis” should be changed to “The analysis”; line 224, “3.3..” should be changed to “3.3.”.
Author Response
REVIEWER 1
The manuscript described the essential oil (EO) of A. sativum cultivated in Peru. According to the GC-Ms, the main components of the EO has been identified to be Diallyl trisulfide which exhibited antioxidant activity against DPPH radical and β-carotene bleaching. Moreover, the computational studies have revealed the best docking score for α-Bisabolol on NADPH oxidase. The manuscript is interesting, and can be accepted after the following minor revision.
Thank you for your comments to improve our manuscript.
Line168-169, Page 4, the authors identified 17 compounds, and four of them are unknown structures. Could the authors identify the specific structure of the four unknown compounds?
R1: Thank you for your comments, in effect we analyze our compounds by GC-MS to show a chemical profile of garlic EO. However, the apparatus could not determine those chemical structures defined as unknown compounds in its library, only we could define its chemical formula. However, according to its formula, they could be sulfur compounds, but we could not infer in those structures.
Line 181-188, page 4, the authors have stated diversity of chemical composition in garlic EO. Will differences in composition lead to differences in application of the garlic oil?
R2: Thank you for your comments, in effect garlic EO showed different chemical compositions according to several factors such as the variety, soil, location, climate and extraction method. Hence, those differences contribute to its biological effect, in our findings the antioxidant activity was major than those EO reported with the same DPPH assay and could be by the presence of Diallyl trisulfide as the main fingerprint in this EO, other EOs had Diallyl disulfide as main phytochemical marker. Although we did not find other in silico studies of garlic as antioxidant, its constituents could modify the activity against determined target linked to the major metabolite.
Some typing mistakes should be revised throughout the manuscript. For example, Line 170, “the analysis” should be changed to “The analysis”; line 224, “3.3..” should be changed to “3.3.”.
R3: Thank you for your comments. They were corrected.
Reviewer 2 Report
The ms "Antioxidant activity, In Silico Study on NADPH Oxidase, and ADMET Prediction of the Chemical Constituents of Allium sativum L. (Garlic) Essential Oil" submitted for publication to Antioxidants descirbes the role of garlic essential oil chemicals as antioxidants.
The ms is in line with journal aims, although some points must be revised:
-the antioxidant activity must be demonstrated in cell line
-ADME properties must be fully analyzed in order to explore a therapeutic effect. Please explain.
.-the interaction with NADPH oxidase must be experimentally demonstrated in vitro.
Author Response
REVIEWER 2
The ms "Antioxidant activity, In Silico Study on NADPH Oxidase, and ADMET Prediction of the Chemical Constituents of Allium sativum L. (Garlic) Essential Oil" submitted for publication to Antioxidants describes the role of garlic essential oil chemicals as antioxidants.
The ms is in line with journal aims, although some points must be revised:
R1: Thank you for your comments to improve our manuscript, which is an original research, which our EO could be used and tested in the future on NADPH oxidase in vitro. Furthermore, we referenced a recent article in which a NADPH oxidase was tested in vitro but they do not explain the chemical interaction between NADPH and sulfur components.
-the antioxidant activity must be demonstrated in cell line
R2: Thank you for your comments. In effect our findings in vitro should be demonstrated in cell lines. However, we used two methods generally those used to measure promissory molecules with antioxidant activity . The β-carotene bleaching assay for evaluating antioxidant activity is one of the common methods used in the field of food chemistry, and DPPH maintain a similar electronic configuration with peroxyl radicals. Otherwise, Other methods are not accessible because they work with aqueous solvents such as ABTS and would be difficult to work with our essential oil. We worked with the cell lines when our laboratory will be provided of those and good conditions post-pandemic.
-ADME properties must be fully analyzed in order to explore a therapeutic effect. Please explain.
R3. A new paragraph was added to understand our findings. They were highlighted in yellow color.
.-the interaction with NADPH oxidase must be experimentally demonstrated in vitro.
R4. In effect, we will work with in vitro models but in this study, we generated a theorical approximation with our docking analysis and molecular dynamic, which showed that only the most active molecule is that with a high energy binding as the alpha-bisabolol. Furthermore, this compound is stable against the other molecules found in the essential oil. Otherwise, alpha-bisabolol can be tested individually and compared with the essential oil .
Thank you for your comments to improve our manuscript.
Reviewer 3 Report
The article is interesting, but it does contain some inaccuracies. They concern:
Title: should be a bit shorter, in its current form it is quite illegible.
This is also related to the suggestion for keywords, which unfortunately are a repeat of the title. In addition, they should be phrases, the most important keys by which the reader can easily find the article in search engines. However, they should not duplicate the title. The authors are asked to correct these inaccuracies.
Chapter: 3. Results and Discussion
It is quite well run.
However, in Table 2, no statistics are available. The authors are asked to indicate the significance of the differences between the individual ingredients for which Antioxidant activities were specified.
Chapter: 4. Conclusions
The authors are asked to formulate 2-3 important conclusions that result from the conducted experiment. Please also indicate the practical potential of the conducted research.
Chapter: References
Although the authors reached for many reports thematically related to the described results of the experiment, it should be clearly stated that these are quite old reports, largely from 15-20 years. In the available international literature there are many more recent reports related to the subject of the manuscript. Therefore, the authors need to refresh the literature a bit, so that it is not only historical, but also indicates the latest reports.
Abstract: "Allium sativum L., also known as garlic, is considered a very important species for its excellent qualities for health and gastronomy ..." - it should be indicated that it is a plant species. It is unnecessary to inform in this part of the article about the type of distillation apparatus used, especially since the authors further inform that the essential oil was obtained by hydrodeistillation.
Next: "Based on these findings, the identification of volatile components of garlic represents the fingerprint of a variety of Peruvian garlic. .." which one could say that mainly the composition of volatile components can be a means of identifying various garlic, and perhaps it is obvious that not only the Peruvian ones.
Introduction: It is quite laconic and contains basic information, generally rather popular science. There are no significant points that make sense of the experiment carried out in relation to the relevant research on garlic around the world. After all, it is a plant that is widely cultivated in many regions of the world and which has received a lot of attention, especially in the last 20 years. Authors are asked to present and locate their research as a whole, not just focusing on local reports. Therefore, it becomes necessary to discuss the influence of various variables, especially site variables, which have a significant impact on the variability of the chemical composition of garlic. When indicating the multitude of varieties, it should be indicated which of them are popular in cultivation and why, as well as their values ​​and diversity in chemical composition, or even in the content of the essential oil itself.
Chapter: 2. Materials and Methods
subsection 2.1. Plant Material - provide references to the analytical procedure in the field of distillation method. After all, this is a standard process as the description suggests. Therefore, it should be stated that the analysis of obtaining the essential oil was performed in accordance with the methodological recommendations, however, the authors do not provide the author of the method. It is understandable that they did not develop this distillation method themselves. Therefore, it is necessary to indicate by whom or what this analytical step was performed according to.
Subsection: 2.3. Antioxidant activity
There is no description of the method modification. The authors must supplement the content of this section with the overall course of analytical proceedings, and in particular with the description of the introduced modification.
Subsection: 2.7. Statistical analysis
Authors must supplement with literature data, indicating the source.
Subsection: 3. Results and Discussion
There is no information about the essential oil content in the tested raw material. It becomes necessary to complete the description of the obtained results of the experiment, because in the introduction the authors indicate the diversity of garlic grown in Peru. Therefore, this type of information, not necessarily in tabular form, should be provided.
Author Response
REVIEWER 3
The article is interesting, but it does contain some inaccuracies. They concern:
Title: should be a bit shorter, in its current form it is quite illegible.
R1: Thank you for your observations: We modified by this: In Vitro Antioxidant Activity, and In Silico Study of the Chemical Constituents of Allium sativum L. (Garlic) Essential Oil on NADPH Oxidase
This is also related to the suggestion for keywords, which unfortunately are a repeat of the title. In addition, they should be phrases, the most important keys by which the reader can easily find the article in search engines. However, they should not duplicate the title. The authors are asked to correct these inaccuracies.
R2: Thank you for your details, we modified the title and keywords.
Chapter: 3. Results and Discussion
It is quite well run.
However, in Table 2, no statistics are available. The authors are asked to indicate the significance of the differences between the individual ingredients for which Antioxidant activities were specified.
R3: Thank you for your observation, we included the statistical analysis.
Chapter: 4. Conclusions
The authors are asked to formulate 2-3 important conclusions that result from the conducted experiment. Please also indicate the practical potential of the conducted research.
R4: Thank you for your information, we added additional paragraphs in the conclusion.
Chapter: References
Although the authors reached for many reports thematically related to the described results of the experiment, it should be clearly stated that these are quite old reports, largely from 15-20 years. In the available international literature there are many more recent reports related to the subject of the manuscript. Therefore, the authors need to refresh the literature a bit, so that it is not only historical, but also indicates the latest reports.
R5: Thank you for your details, we modified old references. However, some references of garlic EOs are older, and we did not found recently papers related to our study with garlic EOs.
Abstract: "Allium sativum L., also known as garlic, is considered a very important species for its excellent qualities for health and gastronomy ..." - it should be indicated that it is a plant species. It is unnecessary to inform in this part of the article about the type of distillation apparatus used, especially since the authors further inform that the essential oil was obtained by hydrodeistillation.
R6: We erased and included this sentence: Allium sativum L., also known as garlic, is a perennial plant widely used as a spice and also considered a medicinal herb since antiquity.
Next: "Based on these findings, the identification of volatile components of garlic represents the fingerprint of a variety of Peruvian garlic. .." which one could say that mainly the composition of volatile components can be a means of identifying various garlic, and perhaps it is obvious that not only the Peruvian ones.
R7: Thank you for your observation, we amended that sentence and include the term one accession of Peruvian garlic.
Introduction: It is quite laconic and contains basic information, generally rather popular science. There are no significant points that make sense of the experiment carried out in relation to the relevant research on garlic around the world. After all, it is a plant that is widely cultivated in many regions of the world and which has received a lot of attention, especially in the last 20 years. Authors are asked to present and locate their research as a whole, not just focusing on local reports. Therefore, it becomes necessary to discuss the influence of various variables, especially site variables, which have a significant impact on the variability of the chemical composition of garlic. When indicating the multitude of varieties, it should be indicated which of them are popular in cultivation and why, as well as their values ​​and diversity in chemical composition, or even in the content of the essential oil itself.
R8: Thank you for your observation, we amended the introduction, however our work shows the importance of garlic EO as a promising antioxidant and that could work on some targets to understand its main mechanism as antioxidant in an organic system, which opens a great possibility to future studies.
Chapter: 2. Materials and Methods
subsection 2.1. Plant Material - provide references to the analytical procedure in the field of distillation method. After all, this is a standard process as the description suggests. Therefore, it should be stated that the analysis of obtaining the essential oil was performed in accordance with the methodological recommendations, however, the authors do not provide the author of the method. It is understandable that they did not develop this distillation method themselves. Therefore, it is necessary to indicate by whom or what this analytical step was performed according to.
R9: Thank you for your observations. We added a reference according to our used method.
Subsection: 2.3. Antioxidant activity
There is no description of the method modification. The authors must supplement the content of this section with the overall course of analytical proceedings, and in particular with the description of the introduced modification.
R2: Thank you for your observations. We added all details of the two methods used in our study.
Subsection: 2.7. Statistical analysis
Authors must supplement with literature data, indicating the source.
R10: Thank you for your observations. We added all details of the two methods used in our study.
Subsection: 3. Results and Discussion
There is no information about the essential oil content in the tested raw material. It becomes necessary to complete the description of the obtained results of the experiment, because in the introduction the authors indicate the diversity of garlic grown in Peru. Therefore, this type of information, not necessarily in tabular form, should be provided.
R11: Thank you for your observations. We added all details in the first paragraph in results section.
Thank you for your comments to improve our manuscript.
Round 2
Reviewer 2 Report
the ms has been revised partially according to reviewer's suggestion.
I think that this ms is now suitable for publication, considering the limitations that authors had due to pandemic.
Reviewer 3 Report
The authors significantly improved the manuscript. They addressed all of the reviewer's suggestions, clarifying the points of concern. The manuscript may be published in its current form.